# Combined Action of Hyper-Harmonized Hydroxylated Fullerene Water Complex and Hyperpolarized Light Leads to Melanoma Cell Reprogramming In Vitro

**DOI:** 10.3390/nano12081331

**Published:** 2022-04-13

**Authors:** Milica Markelić, Dijana Drača, Tamara Krajnović, Zorana Jović, Milica Vuksanović, Djuro Koruga, Sanja Mijatović, Danijela Maksimović-Ivanić

**Affiliations:** 1Faculty of Biology, University of Belgrade, 11000 Belgrade, Serbia; 2Institute for Biological Research “Siniša Stanković”, National Institute of Republic of Serbia, University of Belgrade, 11060 Belgrade, Serbia; dracadiana@gmail.com (D.D.); tamara_krajnovic@yahoo.com (T.K.); sanjamama@ibiss.bg.ac.rs (S.M.); 3TFT Nano Center, 11050 Belgrade, Serbia; zorana.jovic@tftnanocenter.rs (Z.J.); djuro.koruga@gmail.com (D.K.); 4Zepto Hyper Tech, 11070 Belgrade, Serbia; milica.vuksanovic@zepterhypertech.com

**Keywords:** melanoma, water-layered fullerol, hyperpolarized light, senescence, differentiation

## Abstract

(1) Background: Their unique structure and electron deficiency have brought fullerenes into the focus of research in many fields, including medicine. The hyper-harmonized hydroxylated fullerene water complex (3HFWC) formulation has solved the limitations of the poor solubility and bioavailability of fullerenes. To achieve better antitumor activity, 3HFWC was combined with short-term irradiation of cells with hyperpolarized light (HPL) generated by the application of a nanophotonic fullerene filter in a Bioptron^®^ device. The benefits of HPL were confirmed in the microcirculation, wound healing and immunological function. (2) Methods: B16, B16-F10 and A375 melanoma cells were exposed to a wide spectrum of 3HFWC doses and to a single short-term HPL irradiation. (3) Results: Apart from the differences in the redox status and level of invasiveness, the effects of the treatments were quite similar. Decreased viability, morphological alteration, signs of melanocytic differentiation and cellular senescence were observed upon the successful internalization of the nanoquantum substance. (4) Conclusions: Overall, 3HFWC/HPL promoted melanoma cell reprogramming toward a normal phenotype.

## 1. Introduction

Melanoma, a malignant tumor that results from the neoplastic transformation of melanocytes, is considered to be the most aggressive type of skin cancer with a very poor prognosis [1]. Although it accounts for only 5% of all skin malignancies, it is responsible for 50% of deaths from skin cancer [2]. Melanoma incidence and mortality are rising at a worrying rate worldwide and in Europe. According to data from 2018, the estimated incidence rate of melanoma in Europe was 15 cases per 100,000 inhabitants, while the mortality rate was 2.5/100,000 [3]. The latest global cancer statistics show that there were 324,635 new cases and 57,043 deaths from melanoma in 2020, representing 1.7% and 0.6% of all cancer cases and deaths, respectively [4]. Despite the enormous research efforts in cancer therapy and new therapeutic discoveries, the overall success in the treatment of metastatic malignant melanoma is quite limited, so the average survival rate of patients with this disease is estimated at only 20–30 months [5]. Due to the very high mortality rate, lack of efficient therapy and high acquired resistance to known drugs, this type of malignant skin cancer is still a disease that is a challenge to treat, which implies a great need for new therapeutic strategies with minimal side effects. 

Among the diverse spectrum of new products and methods used for therapeutic purposes, the healing power of light has gained significant attention. Photodynamic therapy (PDT) emerged as a modern and rapidly developing method and soon became a promising alternative approach in the treatment of a broad range of diseases, primarily cancer. Over the last 40 years, this therapeutic approach has been used as a non-invasive, non-toxic, well-tolerated and repeatable procedure for tumor treatment. PDT is an approved treatment regime for several cancer types [6]. It was reported to be effective as an alternative treatment for skin cancers, both melanoma and non-melanoma [7,8], even though melanoma is generally considered to be resistant to PDT [9]. The main advantage of this therapeutic approach is its selectivity to malignant, unhealthy tissues. There are at least three known mechanisms through which PDT can induce cancer cell death: direct cellular damage by inducing reactive oxygen species (ROS) production, indirect damage by disrupting the tumor vasculature and stimulation of the immune response against tumor cells by increasing cancer cell-derived antigen presentation to T cells [6,10,11,12].

Photodynamic therapy represents a special type of light therapy that, to a great extent, relies on nanotechnology. It is based on the interaction between light, oxygen and various photosensitizers (PSs), i.e., non-toxic light-sensitive compounds that can be excited by an appropriate light wavelength to induce cancer cells’ death. The combined action of three main elements leads to the generation of cytotoxic ROS which produce irreversible damage to target tissues, causing cell death by necrosis or apoptosis [12,13]. The outcome of PDT depends on the intrinsic properties of PSs which determine their therapeutic efficiency as they can absorb light of a specific wavelength and trigger photochemical or photophysical reactions [14]. More than 1000 natural and synthetic photosensitizing compounds have been discovered thus far [15], some of which are either clinically approved or undergoing clinical trials [16]. The number of new PSs is constantly rising as a result of the persistent pursuit of the ideal photonic and biological properties of these compounds. 

Besides diamond and graphite, fullerenes are the most stable form of carbon, with the molecule C_60_ being the representative member of this carbon-only family. Their unique physical and structural properties and excellent antioxidant effects make fullerenes suitable for biomedical applications. Biological studies of fullerenes and their derivatives started a decade ago and have indicated their growing notability in biology and medicine. The antiviral, antioxidant, anticancer, and antimicrobial actions of fullerenes and their derivatives have been described [17]. Due to fullerenes’ insolubility in a polar solvent and subsequent inapplicability, water-soluble fullerene derivatives have been developed in order to facilitate their potential biomedical uses. Fullerol, C_60_(OH)x, is the first fullerene derivative formed by symmetrical attachment of the 12–48 hydroxyl groups. The free radical scavenging and antioxidative activities of fullerols are their most prominent properties regarding the mechanisms of their activity in biological systems [18]. Their photosensitizing feature is also well exploited in biomedical studies. Upon photoexcitation, fullerols can generate cytotoxic ROS (in that way acting as oxidants [19], which makes them suitable for use in cancer treatment by light-based therapy such as PDT). Many fullerene derivatives with different functional groups attached possessing biological affinity to nearly all types of biomolecules are being investigated for anticancer activity [20,21,22]. 

To overcome the problems of fullerols such as their toxicity at higher concentrations and improve their signaling and influence on biomolecules, the second derivative of fullerenes, hyper-harmonized hydroxylated fullerene water complex (3HFWC), has been patented [23,24]. This nanosubstance is made from C_60_(OH)_24–45_ fullerol, around which water layers are added—(C_60_(OH)_24–45_@(H_2_O)_144–2528_. 3HFWC particles are composed of two main parts—solid state and liquid. The solid phase consists of the C_60_ molecule, covalent OH groups and 3–6 water layers with strong hydrogen bonds and has properties similar to liquid crystalline materials. The liquid phase consists of additional water layers bonded by moderate to weak hydrogen bonds. This nanosubstance is a water-soluble amphiphilic molecule with potential for various biological applications, since the water layers protect biomolecules from the potential toxic effects of C_60_, as well as protecting fullerol from environmental influences [25,26,27]. In this study, the antitumor potential of hyperpolarized light (HPL) and of 3HFWC alone or as a combined treatment was explored on melanoma cells of different invasiveness: B16 and their metastatic counterparts—B16-F10 and human A375 cells. Selection of the cell lines was conducted according to their phenotype characteristics, level of invasiveness and response to therapy, enabling us to delineate the consistency of the effects of the applied treatments depending on tumor grade and metastatic properties. Our results reveal the unexpected behavior of this type of modified fullerol. Oppositely to the mentioned fullerene derivatives whose irradiation triggers ROS-mediated cell death, 3HFWC induced the reprogramming of melanoma cells toward melanocytes accompanied by the development of a senescent phenotype. The process of cancer cell differentiation involves the reversion of their phenotype from a malignant to an original or distant cell type of the same embryonal origin (transdifferentiation) [28,29,30,31,32,33]. This differentiation-based therapeutic strategy has already been utilized in hematological cancers, although its feasibility for solid malignancies is still under consideration, and only a few examples in the literature have been explained thus far [33]. Given that this approach is based on changing rather than killing cells, it may be particularly applicable in the prevention of tumor repopulation triggered by aggressive treatment [34].

## 2. Materials and Methods

### 2.1. Reagents and Cells

Reagents used for cell treatments were: RPMI-1640 culture medium (Biowest, Riverside, MO, USA); fetal calf serum (FCS), phosphate-buffered saline (PBS), trypsin, dimethyl sulfoxide (DMSO), crystal violet (CV), carboxyfluorescein diacetate succinimidyl ester (CFSE), propidium iodide (PI) (all from Sigma–Aldrich, St. Louis, MO, USA); 3-(4,5-Dimethythiazol-2-yl)-2,5-diphenyltetrazolium bromide (MTT), dihydrorhodamine 123 (DHR123; Thermo Fisher Scientific, Waltham, MA, USA); 4-Amino-5-methylamino-2′,7′-difluorofluorescein diacetate (DAF-FM), fluorescein di-β-digalactopyranoside (FDG) (both from Molecular Probes (Eugene, OR, USA)); bovine serum albumin (BSA; AppliChem, St. Louis, MO, USA); paraformaldehyde (PFA; Serva, Heidelberg, Germany); annexin V-fluorescein isothiocyanate (Ann V-FITC) (BD Pharmingen, San Diego, CA, USA); Apostat (R&D Systems, Minneapolis, MN, USA); acridine orange (LaboModerne, Paris, France); penicillin/streptomycin solution (Biological Industries, Cromwell, CT, USA).

Hyper-harmonized hydroxylated fullerene water complex (3HFWC) was synthesized at TFT Nano Center, Belgrade, Serbia, according to the patented procedure [23,24], from hydroxylated fullerene C_60_(OH)_24–45_ (purity 99.99%, Solaris Chem, Vaudreuil-Dorion, Canada) in high-purity water (0.05 µS/cm) under the influence of an oscillatory magnetic field. The maximal water solubility of C_60_(OH)_36_ is 350 g/L; however, a concentration of 1 g/L was used for 3HFWC preparation. To prevent aggregation and agglomeration, fullerol was pre-treated in an ultrasonic apparatus (Stone Age Industries, Powell, WY, USA). The nanosubstance was stored at room temperature (RT). Transmission electron microscopy (TEM) analysis of 3HFWC particles was performed in order to determine the size of their solid state. A drop of 3HFWC solution was applied on a formvar-coated grid and analyzed after drying on a Philips CM12 transmission electron microscope (Philips/FEI, Eindhoven, The Netherlands) equipped with the digital camera SIS MegaView III (Olympus Soft Imaging Solutions, Düsseldorf, Germany). 

Murine B16 and B16-F10 cell lines as well as the A375 human melanoma cell line were purchased from Cell Lines Service GmbH (CLS) (Eppelheim, Germany). Melanoma cells were cultivated in HEPES-buffered RPMI-1640 medium supplemented with 10% heat-inactivated FCS, 2 mM L-glutamine, 0.01% sodium pyruvate and antibiotics (penicillin 100 U/mL and streptomycin 100 μg/mL). Cells were grown in a humidified atmosphere at 37 °C with 5% CO_2_.

Peritoneal exudate cells (PECs) were obtained from C57BL/6 mice by ice-cold PBS lavage of the peritoneum. The cells were left to adhere for 2 h in HEPES-buffered RPMI-1640 medium supplemented with 5% heat-inactivated FCS, 2 mM L-glutamine, 0.01% sodium pyruvate and antibiotics (penicillin 100 U/mL and streptomycin 100 μg/mL). Subsequently, the non-adherent cells were removed by washing with PBS. The handling of animals and the protocol for the isolation of PECs were conducted in accordance with the rules of the European Union and approved by the Institutional Animal Care and Use Committee at the Institute for Biological Research “Siniša Stanković” (IBISS) (No. 02-09/16).

### 2.2. Irradiation Source 

For the irradiation of cells, a Bioptron^®^ 2 device (Bioptron AG, Wollerau, Switzerland) equipped with a nanophotonic fullerene filter was used. This filter, 2 mm thick, was made from polymethyl methacrylate (PMMA) glass with the addition of 0.3% fullerene [35,36]. It allows the production of incoherent, out-of-phase unsynchronized hyperpolarized polychromatic light (HPL) ranging from 400 to 1100 nm. Additionally, the nanophotonic filter material emits low vibrato-rotation energy (0.07–0.20 eV) in the infrared part of spectra (from 5000 to 15,000 nm) with three characteristic peaks at 5811 nm, 8732 nm and 13,300 nm. The light source was adjusted at a distance of 10 cm from the cell dish to provide a uniform spot of 16 cm in diameter with a power intensity of 60 mW/cm^2^ as measured with a spectrophotometer (ILT-350, International Light Technologies, Peabody, MA, USA). The used doses of HPL were: 12 J/cm^2^, 36 J/cm^2^ or 72 J/cm^2^ (after 5, 15 or 30 min treatment). 

In order to determine if irradiation with HPL leads to physical or chemical changes in 3HFWC, the compound was exposed to HPL for 5, 10 and 15 min, and absorption spectra in the UV–visible–infrared (UV–Vis–NIR) (200–900 nm) and Fourier transform infrared (FTIR) domains (2700–3300 nm) (Lambda 900 spectrophotometer, PerkinElmer, Waltham, Massachusetts, MA, USA) were determined.

### 2.3. Viability Assays

For viability assays, melanoma cells were seeded in 96-well plates at a density of 7 × 10^3^, 5 × 10^3^ or 3 × 10^3^ cells per well in 100 µL cultivation media. In order to determine the optimal IC_50_ dose of 3HFWC, cells were incubated with different dilutions of this nanosubstance, corresponding to 0.19–100 µg/mL of C_60_(OH)_24–48_. The treatment lasted 24, 48 or 72 h. To analyze the effects of HPL, cells seeded in sterile plastic Petri dishes were irradiated with HPL at RT for 5, 15 or 30 min at a distance of 10 cm. As an irradiation control, melanoma cells were exposed to daylight at RT for identical periods of time. For the determination of the combined (3HFWC+HPL) effects on melanoma cell viability, 3HFWC treatment (same dose range) was applied for 24 h to B16, B16-F10 and A375 cells followed by HPL irradiation for 5 min, 15 min or 30 min. Viability was measured after 24, 48 or 72 h of incubation by MTT and CV assays. For the MTT assay, the supernatant was removed at the end of the cultivation period and MTT solution (0.5 mg/mL) was added to the cells. Cells were incubated at 37 °C for 30–45 min until formazan crystals were formed. The dye was discarded, and the crystals were dissolved in DMSO. For the CV test, after the supernatant was removed at the end of the cultivation period, cells were fixed with 4% PFA for 10 min at RT. Subsequently, cells were stained with 0.02% CV solution in PBS for 15 min, washed with tap water and air dried, and the dye was dissolved in 33% acetic acid. The absorbance for both MTT and CV tests was measured with an automated microplate reader at 540 nm with a reference wavelength of 670 nm. Cell viability was expressed as a percentage of the untreated control value. All experiments were repeated 3–5 times, and representative data are presented.

In addition, to reveal whether the 3HFWC and HPL treatments affected normal, non-malignant cells’ viability, PECs were seeded in 96-well plates at 2 × 10^5^ density/well, treated with 3HFWC as previously described for melanoma cells and irradiated with HPL for 15 min.

### 2.4. CFSE Proliferation Assay

The standard CFSE staining protocol consists of CFSE staining prior to the experimental treatment. However, in order to prevent CFSE photobleaching by HPL, cells (seeded in 6-well plates at a density of 2 × 10^5^ cells per well in 1 mL cultivation media) were exposed to the IC_50_ dose of 3HFWC for 24 h and then irradiated by HPL/daylight for 15 min. Treatment media were collected and subsequently returned to cell wells after staining with 1 μM CFSE for 10 min at 37 °C. Finally, after the 48 h incubation, cells were trypsinized, dissolved in PBS and analyzed with a CyFlow^®^ Space flow cytometer using the FloMax^®^ software (Sysmex Partec, Goerlitz, Germany), as described previously [25].

### 2.5. Cell Death Assays

For all cell death assays, cells were seeded in 6-well plates at a density of 2 × 10^5^ cells per well in 1 mL cultivation media and subsequently exposed to the IC_50_ dose of 3HFWC and/or HPL treatment (15 min). After 48 h of incubation, cells were stained according to the manufacturer’s procedures. To determine if the experimental treatments induced apoptosis, cells were stained with Ann V-FITC and PI (15 μg/mL) for 15 min at RT. Activation of caspases was analyzed after the treatment with the FITC-conjugated caspase inhibitor Apostat (1 µL/100 µL of 5% FBS in PBS, 30 min at 37 °C). To detect autophagosomes, at the end of the incubation period, cells were stained with acridine orange (10 µM in PBS, 15 min at 37 °C). The fluorescence intensity of the applied fluorophores was analyzed by a CyFlow^®^ Space flow cytometer.

### 2.6. Cell Senescence β-Galactosidase Assay

In order to reveal if 3HFWC and HPL induce cell senescence, detection of senescence-associated β-galactosidase (SA-β-Gal) was conducted. Melanoma cells were seeded in 6-well plates at a density of 2 × 10^5^ cells per well in 1 mL cultivation media and subsequently exposed to the IC_50_ dose of 3HFWC and/or HPL treatment (15 min). After the treatments, cells were stained with 1 mM FDG in deionized water for 1 min at 37 °C. Thereafter, the cell suspension was diluted 10 times in ice-cold cultivation medium. Prior to flow cytometric analysis (CyFlow^®^ Space), the samples were stored at 4 °C.

### 2.7. Detection of Intracellular Production of Reactive Oxygen/Nitric Species (ROS/RNS) and Nitric Monoxide (NO) 

Cells were seeded and treated as described for the cell senescence assay. As for the CFSE staining, the ROS/RNS detection protocol was modified in order to prevent photobleaching by HPL. Cells were treated with the IC_50_ dose of 3HFWC for 24 h and/or irradiated by HPL/daylight for 15 min. Treatment media were collected and subsequently returned to cell wells after staining with 1 μM DHR123 for 20 min at 37 °C. To detect NO production, at the end of the incubation period (48 h), DAF-FM diacetate was added to the 10% FBS-enriched cultivation medium without phenol red (5 µM DAF-FM diacetate, 1 h at 37 °C). To achieve de-esterification, after washing, cells were incubated for 15 min at 37 °C. For both protocols, the fluorescence intensity was measured after 48 h of incubation, by a CyFlow^®^ Space flow cytometer using the FloMax^®^ software. 

### 2.8. Microscopic and Morphometric Analyses 

#### 2.8.1. Transmission Electron Microscopy (TEM)

After the treatment with the IC_50_ dose of 3HFWC for 24 h and/or 15 min HPL irradiation, B16-F10 melanoma cells (seeded in 6-well plates at a density of 2 × 10^5^ cells) were trypsinized and washed in PBS, followed by centrifugation (3 min, 2000 rpm). The cell pellet was fixed in 2.5% glutaraldehyde in 0.1 M Sørensen phosphate buffer (PB). After washing in PB, samples were encased in Bacto-agar, and solidified blocks were cut into 1 mm^3^ cubes. After the postfixation in 1% osmium teroxide, samples were dehydrated using increasing concentrations of ethanol and embedded in Araldite resin (both from Fluka Chemie, Buchs, Switzerland). Prior to ultra-thin sectioning, semi-thin (1 μm thick) sections were obtained using a Leica UC6 ultramicrotome (Leica Microsystems, Mannheim, Germany), stained with toluidine blue and analyzed with a DMLB light microscope (Leica Microsystems). Ultra-thin sections were mounted on grids and counterstained in Leica EM Stain. Sections were examined on a Philips CM12 transmission electron microscope equipped with the digital camera SIS MegaView III (Olympus Soft Imaging Solutions).

#### 2.8.2. Light Microscopy (LM) Examination

For LM examinations, B16, B16-F10, and A375 cells were seeded in 8-chamber slides at a density of 1.5 × 10^4^ cells per chamber and incubated as described. Alternatively, for immunodetection, cells were seeded on cover slips in 24-well plates at 3 × 10^4^ cells per well. At the end of the incubation period, cells were fixed with 4% PFA for 15 min at RT, washed in PBS and stained routinely (hematoxylin eosin, HE) or for the detection of specific cellular constituents: lipids and lipofuscin (Sudan Black B, SBB; Sigma- Aldrich); melanin (Lillie staining); tubulin and myelin basic protein (MBP) (immunocytochemistry). Subsequently, all samples were mounted and examined with a DMLB light microscope (Leica Microsystems). 

For SBB staining of lipids and lipofuscin, fixed cells were incubated in 70% ethanol for 2 min and stained with SBB solution in 70% ethanol (0.7% solution, filtered immediately prior to use) for 8 min. After staining, cells were washed in 50% ethanol and distilled water, mounted and examined. 

For Lillie staining of melanin, fixed cells were incubated in 2.5% ferrous sulphate in distilled water for 1 h. After thorough washing in distilled water, cells were incubated in 1% potassium ferricyanide (dissolved in 1% acidic acid). Following incubation, cells were washed in distilled water and counterstained in 0.1% Nuclear fast red stain solution for 10 min. Cells were mounted and examined. 

For immunocytochemical detection of tubulin and MBP, fixed cells were permeabilized, and endogenous peroxidase blocking followed by protein blocking in 5% FCS at RT for 1 h was performed. Incubation with an adequate concentration of primary antibody (mouse anti- tubulin, polyclonal, 1/100, *v*/*v*, courtesy of Ana Đorđević, Department of Biochemistry, IBISS, Serbia; or mouse monoclonal anti-MBP at 0.5 µg/mL, Biolegend, San Diego, CA, USA) was performed at RT for 1 h. After thorough PBS washing, cells were incubated with anti-mouse secondary antibody in 1% BSA for 30 min, at RT (at a concentration of 1/20 *v*/*v*, Sigma- Aldrich, St. Louis, MO, USA). Following PBS washing, cells were incubated in Extravidin-HRP solution in 1% BSA for 30 min, at RT (concentration 1/20 *v*/*v*, Sigma- Aldrich). Diaminobensidine (DAB) solution with H_2_O_2_ in PBS was prepared according to the manufacturer’s protocol (Dako, Glostrup, Denmark) immediately prior to the incubation. After washing, cells were counterstained with Mayer’s hematoxylin and mounted for microscopic analysis.

For the analysis of nuclear morphology and size, PI staining according to a previously described protocol [25,26] was performed. After mounting, cells were examined with an SP5 confocal microscope (Leica Microsystems).

### 2.9. Statistical Analysis

Analysis of variance (ANOVA), followed by a Bonferroni multiple comparison test or a Student–Newman–Keuls test, was used to determine the significance of the differences between treatments. The statistical significance cut-off point was set at *p* < 0.05.

## 3. Results

### 3.1. HPL Influences the Order of Water Molecules inside 3HFWC without Affecting Hydrogen Bonds

The properties of 3HFWC are presented in Appendix A. As demonstrated by TEM, solid-phase particles of 3HFWC had a size ranging from 5 to 18 nm, and no aggregations nor sediments were noted (even after three years of storage). UV–Vis–NIR spectroscopy revealed limited difference in the absorption of 3HFWC upon irradiation with HPL (Appendix A) without affecting hydrogen bonds, indicating that HPL influenced the dipoles of water molecules around the solid state of 3HFWC, making them more ordered. Additionally, there was no irradiation influence on hydrogen bonds in the FTIR domain (2700–3300 nm). This finding strongly suggests that the reason for the absorbance difference in the samples before and after treatment with HPL lies in the order of water molecules in the liquid phase of 3HFWC. 

### 3.2. 3HFWC and HPL Affect Melanoma Cell Viability In Vitro

To evaluate the effect of the 3HFWC substance and HPL on the viability of melanoma cells, B16, B16-F10, and human A375 cells were exposed to a range of 3HFWC concentrations (corresponding to 0.19–100 µg/mL of C_60_(OH)_24–48_) for 24, 48 and 72 h, to 15 min-long HPL irradiation or to the combination of 3HFWC and HPL. According to the MTT and CV tests, separate treatment with 3HFWC or with HPL did not affect the viability of the tested cell lines (Appendix A). Additionally, neither the separate nor the combined treatments disturbed the viability of the primary PECs (Appendix A), thus confirming their safety in healthy cells. On the other hand, the combined treatment decreased the number of all melanoma cells, with significant effects after 48 h in B16 and B16-F10 cells and 72 h in A375 cells (Figure 1). 

To confirm the viability assays’ results, the number of viable cells was further estimated by counting the nuclei of PI-stained cells (Figure 2A). Microscopic analysis of PI-stained cells and determination of their abundance showed strong inhibition of cell growth in 3HFWC- and 3HFWC+HPL-treated cultures. In addition to the decreased cell abundance, a lower frequency of mitotic figures was found microscopically. Similar effects were demonstrated in all three melanoma cell lines used in this study. In concordance with the lower mitotic rate, CFSE staining of cells revealed strong inhibition of cell division at the same time points. Taken together, it is obvious that the applied treatments slowed down the growth of melanoma cells in vitro (Figure 2B).

The observed discrepancies among the results of the colorimetric (MTT and CV) viability assays, on the one hand, and the microscopic and proliferation capacity analyses, on the other, noted in the 3HFWC-treated cultures, indicate the low accuracy of these assays since they are based on mitochondrial respiration (MTT) or dye binding to DNA/proteins inside the cytoplasm of adherent cells (CV). Both of the parameters do not necessarily reflect cell viability but could also be affected by the alterations in mitochondrial activity and cell size/protein and DNA content. As the PI staining demonstrated, the nuclei of treated cells were more enlightened and enlarged, especially in B16 and B16-F10 cells, thus suggesting alterations in cellular morphology and activity. In addition, no significant increases in cell death by apoptosis, necrosis or autophagy and in caspase activity were noted after these treatments in B16 and B16-F10 cells (Appendix A), confirming the lack of cytotoxic effects of 3HFWC and HPL in these melanoma cells. On the other hand, only a small amount of early/late apoptotic cells were detected upon the treatment of A375 cells (1.5%, 2.5%, 6.2% and 10.1% in control, HPL, 3HFWC and 3HFWC+HPL, respectively), confirming that the inhibition of proliferation is the dominant effect after the treatment. Additionally, prolonged incubation of B16 cells for 96 h confirmed the permanent effect of the nanoquantum substance manifested by the phenotype change and persistent inhibition of proliferation determined by cell viability assessment and light microscopy (Appendix A). Moreover, HPL irradiation supported cell reprogramming triggered by 3HFWC.

### 3.3. 3HFWC and HPL Turn Melanoma Cells to a Senescent State

Morphometric analysis of the tested melanoma cell lines exposed to HPL and 3HFWC alone or in combination revealed that the applied treatments led to an increase in the nuclear size and the appearance of two distinct populations—small, potentially differentiated, and enlarged, resembling senescent cells. More specifically, the senescent- cells were detected as enlarged and flattened with enlightened cytoplasm and nuclei (Figure 3). Accordingly, the increased expression of β-galactosidase in comparison to the untreated controls demonstrated that exposure to 3HFWC and/or HPL triggered cellular senescence (Figure 4A). This effect was noted in all melanoma lines used. In addition, SBB staining and TEM analysis of B16-F10 cells confirmed this finding, since an accumulation of lipid droplets in all treated cultures was observed (Figure 4B). 

Ultrastructural analysis of these cells also revealed mitochondrial alterations in all treated cultures (Figure 4B). More specifically, in contrast to the control cells where the mitochondria were mostly minute, oval or slightly elongated, the mitochondria of cells treated with 3HFWC and 3HFWC+HPL were significantly larger with altered tubular and vesicular cristae. The occurrence of enlarged mitochondria could also be attributed to cell senescence, as previously described in therapy-induced senescent melanoma cells by another group [37]. 

### 3.4. 3HFWC and HPL Stimulate Melanoma Cell Differentiation 

In addition to cell senescence stimulation, the microscopic study demonstrated that 3HFWC and HPL, alone or in combination, also stimulated the transformation of some melanoma cells to a more dendritic phenotype, with smaller, more heterochromatic nuclei (Figure 3B,C). In order to investigate if the dendritic phenotype is a sign of melanocytic or neurogenic Schwann-like cell differentiation, as described previously by us and others [33,38,39,40], the signs of melanogenesis and of the expression of the Schwann cell marker MBP were investigated. Immunocytochemical detection of MBP did not reveal its expression in the treated melanoma cells (Appendix A), thus refuting the neurogenic differentiation of the treated melanoma cells. On the other hand, LM and TEM analysis of melanosomes demonstrated stimulation of melanosome maturation. Mature melanosomes were detected as brown cytoplasmic granules in HE-stained cells of the murine melanoma lines (Figure 5A). 

As described previously, four stages of melanogenesis are detectable by TEM [41]: I—early, non-pigmented endosomes with intraluminal vesicles (ILVs); II—non-pigmented vesicles with parallel premelanosome protein-17 (PMEL17) fibrils inside; III—early pigmented stage with melanin depositions along PMEL17 fibrils; IV—completely pigmented mature melanosome. Our TEM study of murine melanoma cells demonstrated an increased incidence of melanosome stages III and IV in HPL- and 3HFWC-treated cells, thus confirming the stimulation of melanocytic differentiation after the 3HFWC and (to some extent) HPL treatments (Figure 5B). In addition, immunocytochemical detection of tubulin revealed the reorganization of microtubules in these cells, which are visible as bundles inside dendrites (Figure 5C). In many of these cells, melanosomes were visible along the microtubules, while TEM analysis also demonstrated their uptake by endocytosis. Detailed TEM analysis of melanoma cells (B16-F10) treated with 3HFWC demonstrated active internalization of the material which corresponds to 3HFWC according to its size and appearance (Appendix A). Its intake occurred primarily via macropinocytosis. These results strongly indicate that the applied treatments stimulated the differentiation of melanoma cells towards a normal, melanocyte phenotype.

Additionally, in all treated cultures, clusters of small coated vesicles were noted in the cytoplasm or inside endosome-like organelles (Figure 5B). Since no increased incidence of clathrin-dependent endocytosis was detected, it is assumed that these vesicles originated from the Golgi. More specifically, during melanogenesis, coated vesicles with melanosome-destined proteins (including tyrosinase) are transported from the Golgi to endosomes [42,43]. Although this finding demands additional confirmation, it stays in line with our evidence of the melanocytic differentiation of melanoma cells treated with 3HFWC and HPL. 

### 3.5. 3HFWC and HPL Induced NO but Not ROS/RNS Production by Melanoma Cells

Previously, it was reported that ROS/RNS, from oxidative stress, represent the leading molecules responsible for the modulation of the main signaling pathways involved in cellhomeostasis maintenance. Additionally, the same molecules are the major mediators of therapy, promoting different outcomes ranging from inhibited proliferation and phenotype changes to cell death [44]. To evaluate their contribution to the specific mode of action of the applied treatments, endogenous NO as well as hydrogen peroxide and peroxynitrite was detected by redox-sensitive dyes. Regarding NO production, detected by DAF-FM diacetate fluorescence, our results demonstrate different responses among the used melanoma lines. While no detectable effect of the applied treatments was noted in B16 cells, increased NO production was detected in B16-F10 and particularly A375 cell cultures after both the 3HFWC and HPL treatments (Figure 6A). 

DHR123 staining demonstrated that 3HFWC alone or in combination with HPL slightly decreased the production of ROS/RNS in all melanoma cell lines used in this study (Figure 6B). HPL irradiation alone did not affect the production of the analyzed reactive species in the B16 and A375 cell cultures or even slightly increase it in the B16-F10 cell culture. Bearing in mind that HPL alone did not influence the production of ROS/RNS, the moderate scavenging effect observed after the combined treatment could be ascribed to 3HFWC’s antioxidative properties. 

## 4. Discussion

In this study, the antimelanoma action of a new fullerene derivative, 3HFWC, was evaluated. The water layers surrounding the solid-phase fullerene C_60_ and OH groups’ core give this new formula of fullerene properties similar to those of liquid crystalline materials [23,24]. This intervention protected the solid C_60_(OH)_x_ core of 3HFWC from environmental damage and improved the vibration energy transfer of C_60_ onto non-covalent hydrogen bonds of water and biomolecules. Following this hypothesis, the vibration energy transfer from 3HFWC affected the intracellular structures and cellular homeostasis. Until now, the influence of fullerenes on intracellular processes was seen in the context of the cellular redox status, acting as a free radical sponge, on one side, and a ROS producer, on the other [45,46,47]. More specifically, the water layers of polyhydroxylated fullerenes are suggested to scavenge ROS by trapping hydroxyl radicals and converting them to less reactive hydrogen peroxide species [48]. Oppositely, they act as oxidants after illumination with visible light/UV irradiation. Fullerenes are demonstrated to transit to a long-lived triplet excited state, thus transferring energy to molecular oxygen and producing highly reactive singlet oxygen [49,50]. This photosensitizing property of fullerenes makes them promising candidates for the selective anticancer PDT since focused irradiation of tumor lesions leads to localized cell death and tissue damage of the treated lesions [51,52]. In our study, the interplay between HPL and 3HFWC did not result in the compound’s sensitization and subsequent cytotoxicity, but rather in cell reprogramming induction. The presented results clearly show that this delicate modulatory effect on the cell phenotype is triggered by 3HFWC and additionally supported by HPL illumination, which is probably the consequence of the different order of water molecules around the solid state of 3HFWC upon irradiation. 

For the first time, we have demonstrated the strong antitumor effects of 3HFWC on both low- and high-grade murine melanoma cells (B16 and B16-F10, respectively) as well as on invasive human melanoma cells (A375) without disturbing the viability of primary healthy cells. This effect is dominantly driven by the inhibition of proliferation as well as by the establishment of cell senescence and melanocytic differentiation. Senescent and differentiated cancer cells lose their ability to divide but remain viable, as demonstrated previously by our team on melanoma cells [53]. This could explain why the CV assay used in this study did not validate viability inhibition, despite the strong 3HFWC-driven suppression of proliferation in all melanoma cell lines used in this study. As we noted, senescent melanoma cells remained adherent but became enlarged and more translucent, accumulating more lipid droplets, and accordingly binding more color per cell, thus disturbing the linear relation between the color intensity and the number of viable cells. On the other hand, the MTT test demonstrated a less than 30% decrease in formazan production in the B16 and B16-F10 cell lines when 3HFWC was applied for 48 h. This indicates a change in metabolic activity, disturbing the ratio between total cell culture respiration and cell number. This was additionally documented by the visual observation of the often enlarged form of mitochondria upon treatment, with more vesicular and tubular cristae. Real insight into the number of cells after the treatments was obtained by counting the cells, which showed that the decrease in the number of viable cells correlated with a reduced mitotic index. In parallel, redistribution of cells according to nuclear and cellular size after the treatments was observed. In comparison to the control, exposure to 3HFWC+HPL led to an increase in the nuclear size and the appearance of two distinct subpopulations: one resembling potentially differentiated cells, and the other senescent-like cells. Melanoma cell senescence is frequently accompanied by differentiation [53,54], as was shown after the 3HFWC treatment, since morphological signs of transformation toward primary melanocytes are noted: dendrite outgrowth followed by microtubular cytoskeleton remodeling, increased melanin synthesis, melanosome maturation and transportation along the dendrites. These data provide evidence of the strong antiproliferative effect of 3HFWC in both solid and metastatic melanoma cells, demonstrating its ability to revert melanoma cells from a proliferating to a differentiated/senescent state. In addition, our study demonstrated the stimulation of cell senescence and melanocytic differentiation in HPL-irradiated melanoma cells without significant effects on cell proliferation and viability. However, when HPL irradiation was combined with 3HFWC treatment, a significant decrease in both cell viability and proliferation was noted, suggesting their co-action in the stimulation of melanoma cell senescence and melanocytic differentiation. Inhibited division and melanocytic differentiation usually interfere with the decreased action of the PI3K/Akt and MAP kinase signaling pathways, and almost always with the permanent inhibition of the p70S6 kinase and S6 protein [29,30,32]. Prosenescent and differentiation therapies are proposed as new, promising and less aggressive approaches in anticancer therapy as opposed to the conventional cytotoxic chemotherapy and radiotherapy [55,56,57]. It is important to note that, in our study, senescence was obvious in all tested melanoma cell lines despite their initial difference in the redox status as well as the diverse cell-specific production of NO upon all applied treatments. More specifically, the aggressive form A375 and the metastatic cell line B16-F10 showed enhanced endogenous production of NO upon the treatments, while this effect was not visible in the less aggressive B16 clone. The consequences of intracellular NO production are defined by the already established redox balance, leading to different outcomes which are cell-specific [44]. While A375 constitutively expresses the inducible form of NOS, B16-F10 expresses endothelial NOS [58,59]. Apart from the initial difference in their NOS status and production of NO, the outcome of the treatment with 3HFWC and/or HPL was equally efficient in all tested lines, indicating that their mechanism is not explicitly related to NO production. Even though the literature data indicate that the establishment of cell senescence can be connected with NO production in aggressive forms of melanoma, it is clear that, in our study, NO was not a prerequisite for the presence of senescent cells triggered by the treatment [60]. Our results demonstrate the slight antioxidative properties of 3HFWC, since decreased ROS/RNS production was confirmed in all melanoma cell lines used after the treatment with the nanoquantum substance. The same effect on ROS/RNS production was shown after the combined 3HFWC+HPL treatment, while no stimulation of reactive species production was noted after HPL irradiation alone. This excludes the photosensitizing properties of HPL on the 3HFWC nanoquantum substance. On the other hand, the obtained results underline the possibility that irradiation with HPL potentiated 3HFWC activity through molecular structure editing. Fine-tuning of the cellular redox status in response to the treatment can be in the background of the cell reprogramming and explain the absence of cytotoxicity that is always connected with the increased production of reactive species. 

The unusual profile of the scavenged ROS/RNS in parallel with the NO enhancement in B16-F10 and particularly A375 cells opened the possibility of NO transfer to the free SH group of specific cysteine residues, a process known as S-nitrosylation that strongly influences signaling pathways involved in cancer regulation from survival to death [61]. Since it was shown that S-nitrosylation of caspase-3 represents an inhibitory signal for its activation, the absence of cytotoxic effects upon the treatment of 3HFWC and/or HPL is in concordance with the mentioned facts [62]. More specifically, analysis of possible involvement of cell death induction in the antimelanoma effects of 3HFWC and/or HPL did not reveal caspase activation, apoptosis, necrosis or autophagic cell death stimulation, especially in both low- and high-grade murine melanoma cell lines, confirming once more that irradiation of 3HFWC did not result in the classical photodynamic principle but rather in the transformation of the malignant phenotype. The concept of conventional anticancer strategies is, in general, based on the induction of DNA/protein damage, thus killing cancer cells, mostly by caspase-dependant apoptotic pathways. However, cancer cell death frequently leads to compensatory proliferation and subsequent tumor progression [63,64]. Killing-based approaches promote tumor repopulation through therapy-induced cell debris or direct communication between dying and neighboring cells [65,66]. Moreover, debris produced in response to therapy influences the tumor microenvironment by promoting protumorigenic features of its constituents. Accordingly, replacement of killing-based approaches with differentiation-based therapies enables departing from this vicious circle. The induction of differentiation interrupts the communication and delivery of proliferative signals from dead to live cells [33,34]. A recent study demonstrated that no compensatory proliferative response of melanoma cells was elicited after the induction of caspase-independent apoptosis [67]. In this context, treatment with 3HFWC alone or with HPL, oppositely to other photodynamic approaches based on cell death mediated by reactive species, affected the malignant cell phenotype in the nonaggressive mode, leading to maturation/senescence. Several examples of the differentiation of melanoma cells toward melanocytes or distant Schwann-like cells upon exposure to naturally occurring or synthetic compounds have been described [29,30,31,32]. Translation from in vitro to in vivo studies and establishment of an appropriate experimental setting for the investigation of this subject are of essential importance for high-grade tumors with a weak response to conventional treatments and frequent relapses. Summarizing all that has been mentioned, the treatments proposed in this study offer a nonaggressive strategy for tumor growth suppression in vitro, making them worthy of further preclinical evaluation in this and similar pathologies. 

## Figures and Tables

**Figure 1 nanomaterials-12-01331-f001:**
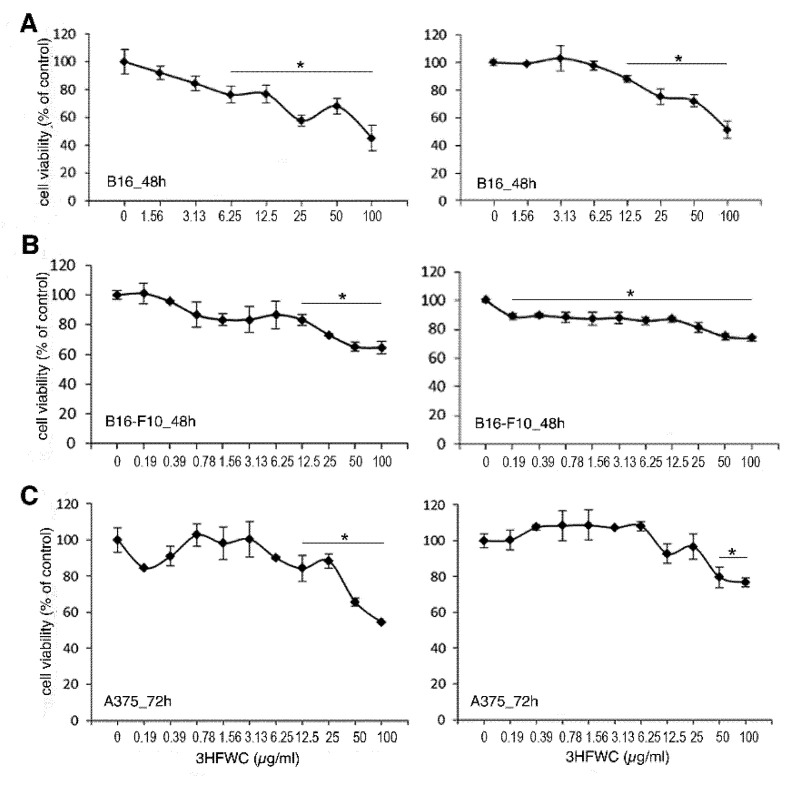
Inhibition of melanoma cell viability by hyper-harmonized hydroxylated fullerene water complex (3HFWC) and hyperpolarized light (HPL) in vitro. (**A**) B16, (**B**) B16-F10 and (**C**) A375 cells were treated with the indicated dose range of 3HFWC (measured as µg/mL of fullerol) and with HPL, for 15 min. The viability was evaluated by MTT (left panel) and crystal violet (CV) (right panel) assays. The results after 48 h (**A**,**B**) and after 72 h of incubation (**C**) are shown. * Significant if *p* < 0.05 in comparison to untreated cells.

**Figure 2 nanomaterials-12-01331-f002:**
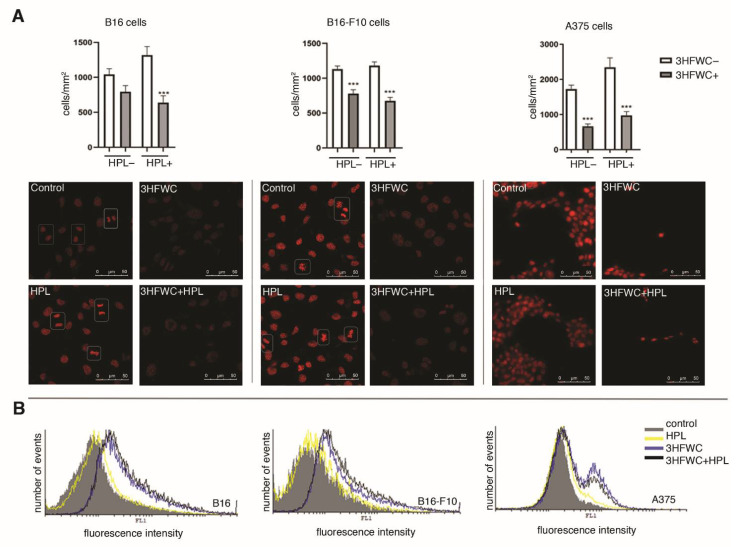
Inhibition of melanoma cell proliferation by 3HFWC and/or HPL in vitro. (**A**) Cell abundance (cells/mm^2^) and propidium iodide staining (white squares—mitotic figures; magnification: ×630, scale bar: 50 µm); (**B**) carboxyfluorescein succinimidyl ester (CFSE) staining; *** significant if *p* < 0.001 in comparison to untreated (control) cells.

**Figure 3 nanomaterials-12-01331-f003:**
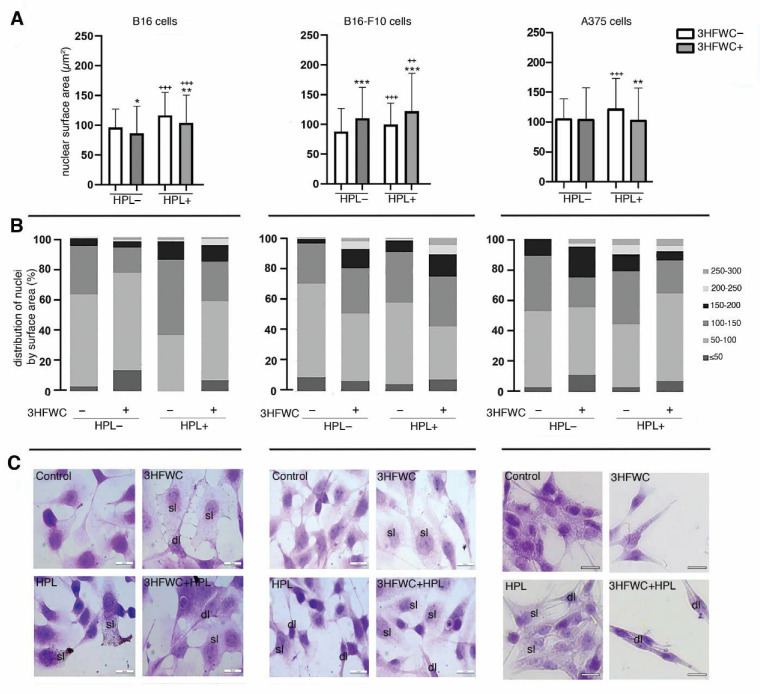
Morphometric and microscopic analysis of melanoma cells’ nuclei after the treatments with 3HFWC and/or HPL. (**A**) Average nuclear surface area (µm^2^); (**B**) distribution of nuclei based on their surface area; (**C**) representative micrographs of hematoxylin and eosin (HE)-stained melanoma cells; sl—senescence-like; dl—dendritic-like phenotype (magnification and scale bar: ×1000, 20 µm). Statistical significance, in comparison to untreated (control) cells: * if *p* < 0.05, ** *p* < 0.01, *** *p* < 0.001; in comparison to identical treatment regarding 3HFWC without HPL irradiation: ^++^ if *p* < 0.01, ^+++^
*p* < 0.001.

**Figure 4 nanomaterials-12-01331-f004:**
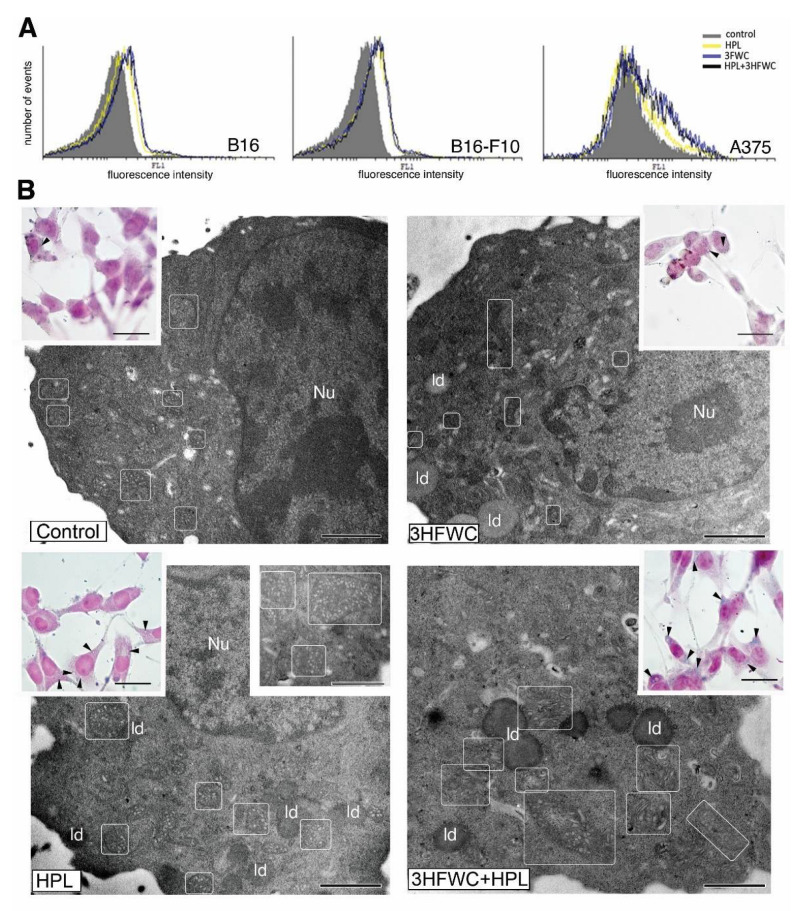
Cell senescence stimulation by 3HFWC and/or HPL. (**A**) Senescence-associated β-galactosidase detection; (**B**) microscopic analysis of senescent phenotype; TEM analysis of B16-F10 cells (Nu—nucleus; ld—lipid droplet; in rectangles—mitochondria) and Sudan black B staining of lipid droplets (inserts, black arrowheads); magnification and scale bar: TEM: ×13,000, 1 µm; LM: ×1000, 20 µm.

**Figure 5 nanomaterials-12-01331-f005:**
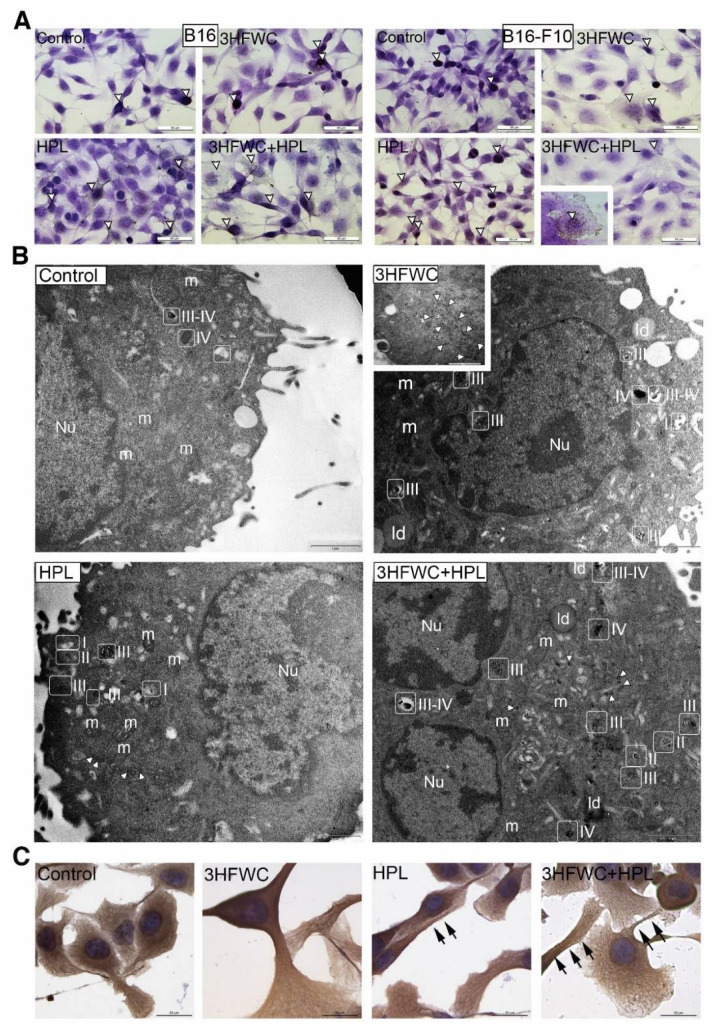
Melanocytic differentiation of B16 and B16-F10 melanoma cells treated with 3HFWC and/or HPL. (**A**) HE staining: occurrence of melanoma cells with brownish mature melanosomes (white triangles); (**B**) TEM: melanosomes in different stages of maturation (I–IV) in B16-F10 cells; small coated vesicles in the cytoplasm (white arrowheads) (Nu—nucleus; m—mitochondrion; ld—lipid droplet); (**C**) immunocytochemical detection of tubulin in B16 cells treated with 3HFWC and HPL; melanosomes along these cytoskeletal elements (black arrows) (magnification and scale bar: A—×400, 50 µm; B—×13,000, 1 µm; C—×1000, 20 µm).

**Figure 6 nanomaterials-12-01331-f006:**
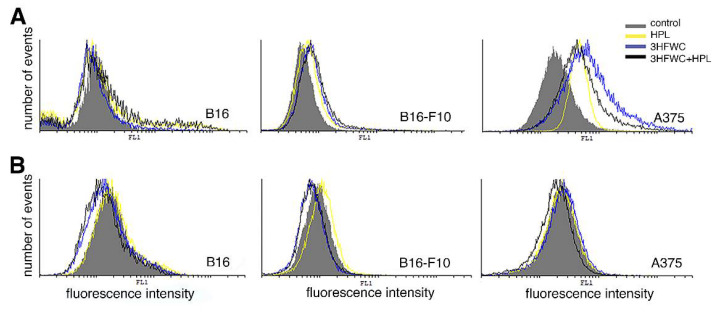
Production of nitric oxide (NO) and reactive oxygen and nitrogen species (ROS/RNS) in melanoma cells after the treatments with 3HFWC and/or HPL. (**A**) DAF-FM diacetate staining; (**B**) dihydrorhodamine 123 (DHR123) staining.

## Data Availability

Data are contained within the article.

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
