# Peer review of "Combined Action of Hyper-Harmonized Hydroxylated Fullerene Water Complex and Hyperpolarized Light Leads to Melanoma Cell Reprogramming In Vitro"

_nanomaterials, 2022, doi:10.3390/nano12081331_

Round 1
Reviewer 1 Report
The manuscript submitted by Dr. Markelic et al. deals with the effect of hyperpolarized light irradiation to hydroxylated fullerenes in vivo, which lead to the death and senescence of melanoma cells. This is an important topic on photodynamic therapy for cancer. Therefore, I think that the paper may contribute to developing the research area of PDT technique. There are however several concerns which the authors must address prior to publication.
The results obtained by morphometric analysis (Figures 3 and 4) are seemed to be ambiguous to show the senescence of the tumor cells. To clarify this, the authors should add the experimental result of viability change in a term longer than 72 h after HPL irradiation.
In Figure S1, what is difference in photos A, B, and C? The character A looks disappeared and C is very unclear.
The size of some of the letters in figures is too small to recognize what those letters mean in Figures 2, 3, and so on.
The specific explanations about the experimental results and figures are wholly too simple to understand. For example, more detailed explanation on Figure 2B is necessary.
In the Discussion section, the first parts in lines 405-432 are tedious, because the content of that part is about the background of the study. If necessary, such general view should move to the Introduction part.
Author Response
The manuscript submitted by Dr. Markelic et al. deals with the effect of hyperpolarized light irradiation to hydroxylated fullerenes in vivo, which lead to the death and senescence of melanoma cells. This is an important topic on photodynamic therapy for cancer. Therefore, I think that the paper may contribute to developing the research area of PDT technique. There are however several concerns which the authors must address prior to publication.
Thank you for your comments. We believe that corrections incorporated in the manuscript significantly improve it. Please find point by point reply below.
The results obtained by morphometric analysis (Figures 3 and 4) are seemed to be ambiguous to show the senescence of the tumor cells. To clarify this, the authors should add the experimental result of viability change in a term longer than 72 h after HPL irradiation.
Although we agree with the peer reviewer that morphometric analysis is not enough to confirm senescence stimulation, according to our opinion combination of different techniques including light and electron microscopy analyses (which have shown increased and enlightened cell appearance with increased nuclear size, mitochondrial alterations, lipids/lipofuscin accumulation) as well as β-Gal detection, enable us to claim that cell senescence is one of the main mechanisms of antitumor activity of 3HFWC and HPL in this study. In addition, our results have demonstrated decreased cell viability upon prolonged incubation (96 h) which is accompanied by decreased proliferative capacity and not by cell death stimulation (now added in the Suppl.material, Figures S5 and S6). These findings also speak in favor of cell senescence stimulation.
In Figure S1, what is difference in photos A, B, and C? The character A looks disappeared and C is very unclear.
We agree with the peer reviewer comments about the Figure S1 (now Figure S8). Therefore, instead of A, B and C characters we have written the names of the treatments. Also, we have added the additional explanation in the figure caption in the Suppl. file. Regarding the question about the differences between the micrographs, we omitted to clarify that our aim only was to present the ultrastructural evidences of 3HFWC uptake in the cultures treated with this nanosubstance, not the differences between the treatments.
The size of some of the letters in figures is too small to recognize what those letters mean in Figures 2, 3, and so on.
The authors are thankful for your comment and have changed the panels and letters size in the figures.
The specific explanations about the experimental results and figures are wholly too simple to understand. For example, more detailed explanation on Figure 2B is necessary.
Thank you for your observation. More detailed explanations are incorporated throughout the text and highlighted in yellow.
In the Discussion section, the first parts in lines 405-432 are tedious, because the content of that part is about the background of the study. If necessary, such general view should move to the Introduction part.
Thank you very much for your criticism. This part is significantly shortened and according to the request of other reviewers the discussion is corrected. All changes are marked in yellow.
Reviewer 2 Report
In this work, the author analyzed the antitumor effect of 3HFWC and/or HPL and found that the combination of 3HFWC and HPL treatment could significantly decrease of both cell viability and proliferation. Notably, the antitumor mechanism was through the production of NO to promote cell senescence rather than traditional ROS to induce cell death. This work offered a nonaggressive manner in tumor growth suppression and was worth further preclinical evaluation in this and similar pathologies. Several questions need to be answered before its publication.
- The characterization of water solubility and morphology of 3HFWC should be given.
- How is the endocytosis efficiency of 3HFWC in the three kinds of selected cells?
- What does the control experiment in Figure 2 refer to? In Figure 1, the viability of the experimental group was lower than the control group, why is the PI staining of the control group darker than that of the experimental group in Figure 2? Is it contradictory?
4. References listed at the end of the manuscript have too many formatting inconsistencies.
Author Response
In this work, the author analyzed the antitumor effect of 3HFWC and/or HPL and found that the combination of 3HFWC and HPL treatment could significantly decrease of both cell viability and proliferation. Notably, the antitumor mechanism was through the production of NO to promote cell senescence rather than traditional ROS to induce cell death. This work offered a nonaggressive manner in tumor growth suppression and was worth further preclinical evaluation in this and similar pathologies. Several questions need to be answered before its publication.
Thank you for your comments. We believe that corrections incorporated in the manuscript significantly improve it. Please find point by point reply below.
The characterization of water solubility and morphology of 3HFWC should be given.
Thank you for your observation. The data about 3HFWC water solubility, concentration and morphology are added to Introduction and M&M sections. Additionally, results of TEM analysis of 3HFWC size, as well of UV-Vis-NIR and FTIR absorbance after HPL irradiation are incorporated in the text as a new chapter in Results section and presented as Figure S1.
How is the endocytosis efficiency of 3HFWC in the three kinds of selected cells?
Thank you for constructive criticism. TEM analysis is done on B16-F10 cells and it has revealed that 3HFWC compound is internalized by melanoma cells via macropinocytosis. Evidence on increased endocytotic activity in the presence of 3HFWC is extended by the additional microscopic analysis, which is incorporated in Figure S8. Since the effect of applied treatment was quite uniform in all three cell lines, implicating that uptake is not cell specific but rather universal, at least for melanomas, only one cell line is selected for detailed TEM investigation.
What does the control experiment in Figure 2 refer to? In Figure 1, the viability of the experimental group was lower than the control group, why is the PI staining of the control group darker than that of the experimental group in Figure 2? Is it contradictory?
Thank you for your observation. The purpose of using the PI staining in our study was to count the number of nuclei and to evaluate their morphology. The intensity of fluorescence correlated with the condensation of chromatin. Weaker staining upon the treatment with 3HFWC-/+HPL in comparison to untreated (control) cells is in concordance with observed senescence of cells, since it involves presence of enlightened and enlarged nuclei. Moreover, it is evident that there were no apoptotic cells with typical condensed chromatin and fragmented nuclei. In addition, AnnexinV/PI staining and caspase activity results that we have added and incorporated as new Figure S5 confirmed that decreased number of cells was preferentially due to inhibited proliferation and induction of senescence/differentiation but not cell death.
References listed at the end of the manuscript have too many formatting inconsistencies.
Thank for your observation. Reference list is checked.
Reviewer 3 Report
In this manuscript, the authors reported a therapeutic strategy using hyper-harmonized hydroxylated fullerene water complex (3HFWC) and hyperpolarized light (HPL) to induce melanoma cells differentiation. It is revealed that 3HFWC exhibits antioxidative properties and induces NO generation, which leads to melanocytic differentiation and tumor growth suppression in vitro. From our point of view, this manuscript can be accepted after major corrections and our suggestions are as followed:
- Based on the title, there should be more information about cell reprogramming in the introduction section, and the related background in discussion section should be concised and put in the introduction section. It is better to emphasize the significance of cell differentiation as cancer therapy.
- It is mentioned that fullerene derivatives could act as both antioxidants and oxidants under certain conditions. What role exactly does 3HFWC play in this manuscript, quenching free radicals or producing NO (which is one kind of ROS actually)? Is there any production of other ROS such as superoxide anion free radical (·O2-), and how does 3HFWC together with HPL affect the redox equilibrium in melanoma cells?
- Are there any physical or chemical changes of 3HFWC after irradiation of HPL? Please provide some relevant data without cell culture, such as UV-vis absorbance change or NMR change of 3HFWC solution.
- As the author claimed, the differentiation/cell reprogramming is better for tumor growth suppression compared to killing-based approaches. What signal pathway or protein change exactly does the differentiation go through, and what is the cell type after tumor cell differentiation?
- The authors mentioned photodynamic therapy multiple times. How to identify if the treatment leads to cell reprogramming or PDT-induced cell apoptosis, and how to explain the cell viability results in MTT assay? Could the author provide some evidence that 3HFWC & HPL treatment do not lead to caspase activation, apoptosis nor cell autophage?
Author Response
In this manuscript, the authors reported a therapeutic strategy using hyper-harmonized hydroxylated fullerene water complex (3HFWC) and hyperpolarized light (HPL) to induce melanoma cells differentiation. It is revealed that 3HFWC exhibits antioxidative properties and induces NO generation, which leads to melanocytic differentiation and tumor growth suppression in vitro. From our point of view, this manuscript can be accepted after major corrections and our suggestions are as followed:
Thank you for your comments. We believe that corrections incorporated in the manuscript significantly improve it. Please find point by point reply below.
Based on the title, there should be more information about cell reprogramming in the introduction section, and the related background in discussion section should be concised and put in the introduction section. It is better to emphasize the significance of cell differentiation as cancer therapy.
Thank you for your suggestion. The text is modified according to your request. New parts in the Introduction and Discussion sections dealing with cell reprogram with supporting references are incorporated and highlighted yellow.
It is mentioned that fullerene derivatives could act as both antioxidants and oxidants under certain conditions. What role exactly does 3HFWC play in this manuscript, quenching free radicals or producing NO (which is one kind of ROS actually)? Is there any production of other ROS such as superoxide anion free radical (·O2-), and how does 3HFWC together with HPL affect the redox equilibrium in melanoma cells?
Thank you for your constructive criticism. We agree with your statement about dual behavior of fullerene derivatives in different circumstances, especially with the fact that the main way of their action is related to reactive species scavenging or production. However, 3HFWC substance is qualitatively different from other molecules of C60 family and its physical/chemical characteristics are now incorporated in the text and ESI. Accordingly, behavior of 3HFWC is significantly different from its solid core. We observed that 3HFWC scavenging potential is moderate while the impact on NO in iNOS+ A375 cells, and eNOS+ B16-F10 is rather connected with 3HFWC influence on NOS enzymes, than NO production per se. The fact that compound showed similar activity in all three cell lines with different NO status, underlines that 3HFWC pivotal mode of action is not dependent preferentially on this molecule, but also didn’t exclude the contribution of elevated NO level on physiology of the cells where it is produced. In A375 cells where endogenous NO is necessary factor for their growth, the change of NO production upon the treatment was the most obvious. To define its contribution to 3HFWC activity in this cell line, we designed the experiment with co-treatment with aminoguanidine (AG), a known inhibitor of iNOS. Please, see Figure X in attached pdf file.
In concordance with our previous results, 3HFWC alone provoked preferentially differentiation of cells, since cells with primarily dendritic phenotype are seen microscopically. HPL on the other hand, stimulated senescence of the cells, as enlightened cells with enlarged nuclei were seen (Figure X). The obtained results showed strong cytotoxic effect of AG, as well senescence of survived population, when applied alone or upon irradiation with HPL. In combination with 3HFWC, potentiated AG cytotoxic action was noted, indicating indirectly that 3HFWC affected NO production by iNOS and/or that 3HFWC realizes its antitumor activity through alternative pathway. This of course requires more detailed analysis and since it is out of the scope of this study, we didn’t include it in the manuscript.
As you have suggested, we have also measured the production of superoxide anion by dihydroethidium (DHE) staining of melanoma cells. As seen on Figure Y (in attached pdf), there is no change in its production, so we also present it just here as a part of the response to your request.
Are there any physical or chemical changes of 3HFWC after irradiation of HPL? Please provide some relevant data without cell culture, such as UV-vis absorbance change or NMR change of 3HFWC solution.
Thank you for your observation. The results representing the size of 3HFWC, UV-Vis-NIR and FTIR after HPL irradiation are incorporated in the text, as well as in the supplementary figure (Figure S1).
As the author claimed, the differentiation/cell reprogramming is better for tumor growth suppression compared to killing-based approaches. What signal pathway or protein change exactly does the differentiation go through, and what is the cell type after tumor cell differentiation?
The authors are thankful for your comment. In addition to the description of signs of differentiation towards melanocyte phenotype, which are already given in the Results section, further explanation was elaborated in the Discussion section.
The authors mentioned photodynamic therapy multiple times. How to identify if the treatment leads to cell reprogramming or PDT-induced cell apoptosis, and how to explain the cell viability results in MTT assay? Could the author provide some evidence that 3HFWC & HPL treatment do not lead to caspase activation, apoptosis nor cell autophage?
Thank you for your suggestion. A new figure showing apoptosis/caspase activity/autophagy (Figure S5) is incorporated as well as additional explanation throughout the text.
Round 2
Reviewer 3 Report
I am satisfied with the revisions the authors have made. This manuscript can be accepted at present form.